# New Fossil Evidence Suggests That Angiosperms Flourished in the Middle Jurassic

**DOI:** 10.3390/life13030819

**Published:** 2023-03-17

**Authors:** Lei Han, Ya Zhao, Ming Zhao, Jie Sun, Bainian Sun, Xin Wang

**Affiliations:** 1Key Laboratory of Minerals Resources in Western China (Gansu Province), School of Earth Sciences, Lanzhou University, Lanzhou 730000, China; 2Ningxia Geological Museum, 301 Eastern People’s Square Street, Yinchuan 750000, China; 3Shaanxi Key Laboratory of Early Life and Environments, State Key Laboratory of Continental Dynamics, Department of Geology, Northwest University, Xi’an 710069, China; 4State Key Laboratory of Palaeobiology and Stratigraphy, Nanjing Institute of Geology and Palaeontology, CAS Center for Excellence in Life and Paleoenvironment, Chinese Academy of Sciences, Nanjing 210008, China

**Keywords:** angiosperms, Middle Jurassic, Northwest China, Micro-CT, *Qingganninginfructus*

## Abstract

Angiosperms are a group of plants with the highest rate of evolution, the largest number of species, the widest distribution and the strongest adaptability. Needless to say, angiosperms are the most important group for the humans. The studies on the origin, evolution and systematics of angiosperms have been the major challenges in plant sciences. However, the origin and early history of angiosperms remains poorly understood and controversial among paleobotanists. Some paleobotanists insist that there were no angiosperms in the pre-Cretaceous age. However, this conclusion is facing increasing challenges from fossil evidence, especially Early Jurassic *Nanjinganthus*, which is based on over two hundred specimens of fossil flowers. Studying more fossil plants is the only reliable way to elucidate the origin and early evolution of angiosperms. Here, we document a new species of angiosperms, *Qingganninginfructus formosa* gen. et sp. nov, and provide the first detailed three-dimensional morphology of *Qingganninginfructus* gen. nov from the Middle Jurassic of Northwest China. A Micro-CT examination shows that the best-preserved fossil infructescence has eleven samaroid fruits, each with a single basal ovule. Since these fossils are distinct in morphology and organization from all organs of known gymnosperms and angiosperms (the latter are defined by their enclosed ovules), we interpret *Qingganninginfructus* as a new genus of angiosperms including a new species, *Q. formosa* gen. et sp. nov., and an unspecified species from the Middle Jurassic of Northwest China. The discovery of this new genus of angiosperms from the Middle Jurassic, in addition to the existing records, undermines the “no angiosperms until the Cretaceous” stereotype and updates the perspective on the origin and early history of angiosperms.

## 1. Introduction

Angiosperms are the most diversified and widely distributed plant group in the current world. However, the origin and early history of angiosperms are poorly understood and controversial. Although paleobotanists have investigated much effort and applied various technologies to decipher these mysteries, and despite the increasing fossil evidence from the Jurassic and Early Cretaceous shedding new light on the origin and early evolution of angiosperms, the rarity of fossil specimens of early angiosperms [1,2] undermines the robustness of the conclusions concerned with early angiosperms [3,4,5]. For example, repeated reports of angiosperm traces from the Jurassic and Triassic by Cornet [6,7,8,9,10], Hochuli and Feist-Burkhardt [11,12], Wang and his colleagues [13,14,15,16,17,18,19,20,21,22,23,24] were largely ignored or downplayed. This difficulty cannot be overcome simply by number of specimens, as the situation appears not obviously improved since *Schmeissneria* [19,20] and *Nanjinganthus* [13,14,15], both based on over tens or hundreds of specimens, were reported from the Early Jurassic: the “No angiosperms until the Cretaceous” stereotype adopted by many paleobotanists [25,26] and taught in classrooms over the past six decades remains intact. Underlying these controversies is the lack of a consensus on the criterion for fossil angiosperms among botanists [14,27]. Previously, many paleobotanists documented early angiosperms without giving their applied criteria for fossil angiosperms. The situation began to improve when paleobotanists began discussing the criterion explicitly [14,19,20,28,29,30,31,32]; however, a side effect of such discussion is that different and conflicting criteria raised by various authors triggered wrangles among botanists [14,27]. Therefore, both fossil evidence and a clearly declared practical criterion for fossil angiosperms are required to expel the stereotype [17,18].

Micro-CT is a newly available non-destructive detecting technology that allows us to envision internal features of plants or fossilized organisms [33]. Using this new technology, Fu et al. [15] has recently revealed the ovules enclosed in integral and closed ovaries of *Nanjinganthus*. This successful attempt encouraged us to use Micro-CT to reexamine some previously reported fossil plants named as *Drepanolepis formosa* by Zhang et al. [34]. The result indicates that these fossil plants from the Middle Jurassic of Northwest China actually have fruits with enclosed ovules/seeds. Since angio-ovuly is a feature so far restricted only to angiosperms [27,28,29], we are therefore confident to interpret *Qingganninginfructus* as a new genus of angiosperms. Significantly, this new discovery does not only confirm the existence of angiosperms in Jurassic, but it also suggests that angiosperms could have flourished in the Jurassic, considering the fossil abundance of the genus in our examined strata. It also suggests that at least some angiosperms might have been wronged as gymnosperms previously, due to lack of clear criteria for fossil angiosperms and practical detecting technologies. Therefore, it is necessary to re-scrutinize some similar fossils of seed plants in museum collections.

## 2. Materials and Methods

The present fossil plants were collected from the Middle Jurassic Shimengou Formation in Delingha, Qinghai (97°40′ E, 36°57′ N), Yaojie Formation in the Yaojie Basin in central Gansu (102°50′ E, 36°25′ N), and Yan’an Formation in Lingwu, Ningxia (106°40′ E, 38°4′ N) (Appendix A). The type locality, Yaojie Basin, one of the main coal-bearing basins in Gansu Province, belongs physiognomically to Qilian Mountain. The Middle Jurassic Yaojie Formation overlies the Lower Jurassic Tandonggou Formation and underlies the Upper Jurassic Xiangtang Formation. It represents the major coal-bearing strata in the basin. The Yaojie Formation is divided into five members in an ascending order: conglomerate member (J_2_y^1^), coal-bearing member (J_2_y^2^), mudstone member (J_2_y^3^), oil shale member (J_2_y^4^), and sandy mudstone member (J_2_y^5^). Fossil plants were preserved in all these members. The specimens in this study were collected from the mudstone member, coal-bearing member and sandy mudstone member (Appendix A). The Yaojie Formation has yielded various other fossil plants, including *Equisetites lateralis*, *Neocalamites hoerensis*, *Gansuphyllites multinervis*, *Radicites* sp., *Rhizomopteris yaojieensis*, *Coniopteris bella*, *C. burejensis*, *C. hymenophylloides*, *C. margaretae*, *C. lanzhouensis*, *C. simplex*, *C. spectabilis*, *Eboracia lobifolia*, *Gonatosorus shansiensis*, *Clathropteris* sp., *Cladophlebis whitbiensis* var. *punctata*, *Pterophyllum subaequale*, *P.* sp., *Anomozamites* sp., *Tyrmia nathorsti*, *Nilssoniopteris inconstans*, *N. vittata*, *N. tenuinervis*, *N. ingens*, *Nilssonia* cf. *mosseraiyi*, *N. orientalis*, *Hausmannia ussuriensis*, *Ginkgo huttonii*, *G. digitata*, *Ginkgoites chilinense*, *G. sibiricus*, *G. obrutschewi*, *G. elegans*, *G. biformis*, *G. yaojieensis*, *Baiera furcata*, *B. gracilis*, *B.* cf. *guilhaumati*, *B.* sp., *Sphenobaiera* sp., *Phoenicopsis angustifolia*, *P. speciosa*, *Czekanowskia setacea*, *C.* sp., *Ixostrobus lepidus*, *Schzolepis* sp., *Podozamites lanceolatus*, *P.* sp., *Elatocladus* sp., *Brachyphyllum obesum*, *B.* sp., *Ferganiella paucinervis*, *F.* cf. *podozamites*, *Pityostrobus* sp., *Solenites vimineus*, *S. murrayana*, *Elatides* cf. *thomasii*, *E.* sp. *Podozamites lanceolatus*, and *P.* sp., *Carpolithus* sp., resembling typical Middle Jurassic floras in China [35,36,37,38,39,40,41]. The Yaojie flora is similar to the well-known Middle Jurassic floras in China and other countries, such as Yan’an flora and Yorkshire flora, which are dominated by ferns and gymnosperms. The fossil flora and palynological assemblages suggest that the Yaojie Formation is equivalent to the Aalenian-Bajocian (174.1–168.3 Ma) [34,41].

The fossils specimens described here were imaged using a FinePix HS50EXR digital camera and a Leica MZ 12.5 stereomicroscope at Key Laboratory of Mineral Resources in Western China (Gansu Province). Four specimens (LZP-2018-02A, LZP-2018-03B, LZP-2018-06A and LZP-2018-05A) were investigated on a Phoenix v|tome|x m scanner using a voltage of 170 Kv and a current of 200 μA at the State Key Laboratory of Continental Dynamics (Northwest University), Xi’an, Shaanxi Province, China. The specimen was immobilized by wrapping in self-adhesive tape and mounted in a foam block. In total, 1758, 1999, 1998, and 1997 projections were obtained, respectively. The dataset had a resolution between 36.56 and 79.47 μm. Three-dimensional reconstructions were produced by using VG Studio 3.2. All the figures were organized using Photoshop 7.0. All fossil specimens are deposited in the Paleontology Laboratory of the School of Earth Sciences, Lanzhou University, China (LZP-1984-57, LZP-1984-58, LZP-1984-59, LZP-2018-01A, LZP-2018-01B, LZP-2018-02A, LZP-2018-02B, LZP-2018-03A, LZP-2018-03B, LZP-2018-04, LZP-2018-05A, LZP-2018-05B, LZP-2018-06A, LZP-2018-06B, LZP-2018-07A, LZP-2018-07B, LZP-2018-08, LZP-2018-09, LZP-2018-10, LZP-2018-11A and LZP-2018-11B), and Ningxia Geological Museum, Yinchuan, China (GSW615, GSW870).

## 3. Systematic Paleobotany

Genus *Qingganninginfructus* Wang and Sun, gen. nov.

**Generic diagnosis**: Infructescence including numerous samaroid fruits spirally arranged along an elongated rachis. Fruit single or in pairs, bulged basally, elongated or round-triangular, longitudinally striated. Ovule single, basally fixed, anatropous, bitegmic.

**Type species**: *Qingganninginfructus formosa* Wang and Sun, gen. et sp. nov.

**Etymology**: *Qingganninginfructus,* for infructescence (-*infructus*) from Qinghai (*Qing*-), Gansu (-*gan*-) and Ningxia (-*ning*-) in Northwest China.

**Stratigraphic horizon**: the Shimengou Formation in Delingha, Qinghai; the Yaojie Formation in Yaojie, Gansu; and the Yan’an Formation in Lingwu, Ningxia, all in Northwest China (Appendix A).

**Age**: the Aalenian-Bajocian, Middle Jurassic (>168 Ma).

**Type locality**: Yaojie, Gansu Province, China.

**Type horizon**: the Yaojie Formation, Middle Jurassic.

**Remarks**: Part of the specimens studied here were previously placed in *Drepanolepis formosa* by Zhang et al. [34]. This placement was at odds with the documentation of *Drepanolepis* by Schweitzer and Kirchner in 1996 [42], as the fossils figured by Zhang et al. [34] apparently lacked “unten verwachsener, oben freier Deckschuppe” (basal fused, apically free bract) (Tafel 4. Figure 12; Abb. 10b-c) [42]. This discrepancy made our investigation on these specimens necessary. The result indicates that these specimens are not *Drepanolepis* as the ovule/seed is not exposed on the adaxial of the lateral unit. Therefore, we propose a new genus, *Qingganninginfructus* Wang and Sun, for these specimens. We cannot exclude the possibility that future studies on Nathorst’s materials of *Drepanolepis* may find enclosed in situ seeds. If indeed so, at least some *Drepanolepis sensu* Nathorst would represent fossil angiosperms and that angiosperms would have spread widely in Eurasia in the Jurassic and Cretaceous.

Anatropous ovules seen in our fossils are of special interest, as they are more frequently seen in angiosperms, especially when bitegmic. Anatropous ovules have been seen in two Mesozoic taxa, *Drepanolepis* [42] and *Combina* [43]. However, distinctions exist between our fossils and *Drepanolepis* [42]: the ovules in our fossils are enclosed, while the ovules in *Drepanolepis* are exposed. The ovules in *Combina* [43] are anatropous and almost fully enclosed; however, there is no information on the outer integument in *Combina*, and the whole organ’s profile and arrangement of lateral appendages are distinct from those of *Qingganninginfructus* documented here.

The fruit profile of *Qingganninginfructus* conjures to the samaras of *Acer*. However, radiating/longitudinal-appearing venation and lack of reticulate venation in samara’s skewed wing [44,45] prevents us from further comparison in detail.

Among extant angiosperms, *Illicium* (Schisandraceae) and *Euptelea* (Eupteleaceae) are more or less similar to *Qingganninginfructus* in certain aspects. *Illicium* shares separate fruits with a basal anatropous ovule with *Qingganninginfructus*. However, the differences between them are obvious: *Illicium* has fruits arranged in a whorl that have ventral sutures and pointed tips [46], while *Qingganninginfructus* has fruits that are spirally arranged and have truncated tips. Furthermore, *Qingganninginfructus* appears to have no traces of stamen, while flowers of *Illicium* are bisexual [46]. *Euptelea* shares a fruit with a single ovule with *Qingganninginfructus*. However, the differences between them are obvious: a fruit of *Euptelea* has a long slender pedicel, 1–3 apical ovules/seeds, and a more or less enlarged tip [47], while a fruit of *Qingganninginfructus* has a short rachis, 1 basal ovule, and a narrowing or truncated tip.

The occurrence of two integuments in our fossils suggests that the oval body within the fruit most likely is an ovule. This fact suggests that *Qingganninginfructus* has an enclosed ovule (angio-ovuly) and therefore a *bona fide* angiosperm. Although no embryo can be discerned in our specimens, we cannot exclude the possibility that oval bodies enclosed in some fruits may have matured as a seed. To be conservative and simple in description, we prefer the term “ovule” (rather than “seed”) in our description. Correspondingly, “fruit” and “carpel” are also two terms hard to distinguish in our materials and description.

*Qingganninginfructus formosa* Wang and Sun, gen. et sp. nov.

(Figure 1, Figure 2, Figure 3, Figure 4 and Appendix A)

**Synonym**: *Drepanolepis formosa* Zhang 1998, MP-93979, MP-93980, page 280; Plate 50, Figure 1; Plate 51, Figure 1 and Figure 2; Plate 53, Figure 2.

**Specific diagnosis**: In addition to the generic diagnosis, fruit elongated or round-triangular, truncated distally.

**Description**: The infructescence is up to 89 mm long and 29 mm wide (Figure 1a,b, Figure 2a, Appendix A). The rachis is up to 85 mm long, 1.3–2.5 mm in diameter (Figure 1a,b, Figure 2a, Appendix A). Fruits are paired or single (Figure 1a,b, Figure 2a, Appendix A). A fruit includes a fruit proper and a pedicel (Figure 2b,c and Appendix A). The pedicel is approximately 2 mm long, 0.5–0.7 mm in diameter, sheathed (Figure 1b, Figure 2a–c and Appendix A). The fruit proper is 5.9–21 mm long, 2.4–8.6 mm wide, elongated or round-triangular in shape, with longitudinal striations (Figure 1a–c,g, Figure 2h–n, Figure 3a–i, Figure 4a–e, Appendix A). A fruit tip is round or truncated, curving to one side distally (Figure 1a–c,g, Figure 2h–o, Figure 3a,f, Figure 4a, Appendix A). A single ovule is enclosed in each fruit, basally fixed, up to 5.8 × 7 mm, with a 0.63 mm-thick seed coat that has radiating sculpture (Figure 1a,b,e–g, Figure 2a–n, Figure 3a–i, Figure 4a–e, Appendix A). The ovule is anatropous and bitegmic (Figure 3a,b,e).

**Etymology***: formosa* is conserved from the former name of the fossil, *Drepanolepis formosa* Zhang et al., 1998 [34].

**Holotype**: LZP-2018-03A, LZP-2018-03B (Figure 3f–i and Appendix A).

**Further specimens**: LZP-1984-57, LZP-1984-58, LZP-1984-59, LZP-2018-01A, LZP-2018-01B, LZP-2018-02A, LZP-2018-02B, LZP-2018-04, LZP-2018-05A, LZP-2018-05B, LZP-2018-06A, LZP-2018-06B, LZP-2018-07A, LZP-2018-07B, LZP-2018-08, LZP-2018-09, LZP-2018-10, LZP-2018-11A, LZP-2018-011B (Figure 1a–h, Figure 2a–o, Figure 3a–e, Figure 4a–e, Appendix A).

**Depository**: Paleontology Laboratory of the School of Earth Sciences, Lanzhou University, China.

**Remarks**: An associated branch has spines (Figure 2f) and a central canal, which is also seen in the rachis of the infructescence (Figure 2e).

*Qingganninginfructus* sp.

(Appendix A)

**Specific diagnosis**: In addition to the generic diagnosis, fruit elongated and round-tipped.

**Description**: The infructescence is up to 60 mm long and 37 mm wide (Appendix A). The fruits are spirally arranged, 22–30 mm long, 10–12 mm wide, cuneate-shaped and round-tipped (Appendix A). A single ovule 5–7 × 7–9 mm is enclosed in each fruit (Appendix A).

**Depository**: Ningxia Geological Museum, Yinchuan, China.

**Remarks**: As it is represented by only one specimen and it has a fruit profile distinct from the type species, we tend to put it as an unspecified species in *Qingganninginfructus*.

## 4. Discussion

Many controversies in the study of early angiosperms can be attributed to the lack of clear consensus on the criterion identifying fossil angiosperms. Many earlier works on early angiosperms did not give the criteria they applied. Wang [4] and Wang et al. [8] are among a few paleobotanists who explicitly declared their criteria. A new trend in paleobotany is that there are increasing number of botanists joining the group discussing criteria identifying fossil angiosperms [26,30,32]. However, the raised criteria are conflicting each other: Herendeen et al. [26] thought that several characters were “unique angiosperm features”, Sokoloff et al. [30] focused their attention on pentamery of flowers, while Bateman [32] preferred double fertilization and closed carpel. The criteria [26,30,32] are impractical in paleobotany [14]. For example, the criterion raised by Herendeen et al. [26], if adopted, would reject the angiospermous affinity of *Monetianthus* [48], which Herendeen et al. thought to be an angiosperm. It is noteworthy that this criterion was rejected by Friis et al. [49]. Pentamerous flower is a synapomorphy restricted to a subset of angiosperms and should not be taken as a criterion for all angiosperms, otherwise monocots and many others would not be angiosperms. Tomlinson and Takaso [50] found that some conifers also seclude their seeds (although such a seclusion occurs only after pollination) and stated that “angiospermy” is not a feature idiosyncratic of angiosperms; instead, “ovules enclosed before pollination” is a stricter criterion for angiosperms. This criterion has been widely adopted and practiced by various authors [1,13,14,15,16,17,18,23,24,28,29,51,52,53]. The present authors prefer to adopt this criterion to identify angiosperms.

We identified our specimens as angiosperms based on the above criterion and the following observations. **First**, Micro-CT images and videos show that the existence of ovules inside the fruits of *Qingganninginfructus formosa* (Figure 2h–n, Figure 3b–e,g–i and Figure 4b–e). This suggests that the ovules are enclosed, at least when viewed from one side. This observation cannot guarantee that *Qingganninginfructus* is an angiosperm, as contemporary *Schizolepis* may have its basal seeds eclipsed if viewed from the abaxial side. Fortunately, as suggested by its nomenclature, the lateral appendages of *Schizolepis* usually have at least two lobes [54], while the fruits in *Qingganninginfructus formosa* are either single (Figure 1b,c, Figure 2a and Figure 4a) or in pairs (Figure 2c, Figure 3f, Figure 4a and Appendix A). The ovule is assumed to be unitegmic in *Schizolepis*, while the ovule is bitegmic in *Qingganninginfructus formosa* (Figure 3b,e). In addition, the lateral appendages of *Schizolepis* usually have abaxial bracts [54], while the pedicels of fruits in *Qingganninginfructus formosa* appear to be sheathed (Figure 2b). These differences are enough to exclude *Schizolepis* from our further consideration. **Second**, various views of broken fruits indicate that the ovules are sandwiched between two layers of fruit walls in *Qingganninginfructus formosa* (Figure 1c–h, Figure 2d,h–n, Appendix A). On one hand, Micro-CT longitudinal virtual sections of fruits indicate that the ovules are indeed inside the fruit (Figure 4d,e). On the other hand, a cross view of the basal portion of a fruit indicates that the ovule is sandwiched between two layers of fruit walls (Figure 1d and Appendix A). The more-or-less three-dimensional preservation of *Qingganninginfructus* sp. allows us to confirm the above conclusion: in all views of the fruits, the ovules are restricted in the fruits (Figure 1a–h, Figure 2a–n, Figure 3a–i, Figure 4a–e, Appendix A). These observations support the existence of an enclosed ovule in *Qingganninginfructus* and thus its angiospermous affinity, according to the above adopted criterion of angiosperms. This conclusion is not at odds with the conclusion on affinity given by the previous authors, since Zhang et al. [34] wisely left the final placement of the currently studied materials open.

*Qingganninginfructus* is found in the Shimengou Formation in Qinghai, the Yaojie Formation in Gansu, and the Yan’an Formation in Ningxia, all in Northwest China. They are associated with various taxa in Ginkgopsida, Filicopsida, Coniferopsida and Cycadopsida, of the *Coniopteris-Phoenicopsis* flora assemblage typical of the Middle Jurassic [35]. Stratigraphy suggests an Aalenian-Bajocian (Middle Jurassic) age for *Qingganninginfructus*. This age appears incompatible with *Qingganninginfructus*’ angiosperm affinity as angiosperms were previously thought restricted to the Cretaceous and later age [25,26]. However, it is in line with other studies of early angiosperms. For example, Hochuli and Feisthardt [11,12] have found pollen grains hard to distinguish from those of angiosperms in the Triassic; Fu et al. [13,14,15] have found over two hundred flowers from the Lower Jurassic in Nanjing, Jiangsu Province, China; Han et al. [24] have found an herbaceous angiosperm in the Middle-Upper Jurassic of Inner Mongolia, China; Liu and Wang [17] have documented a perfect flower from the Middle Jurassic of western Liaoning, China. In addition, other independent studies also favor a pre-Cretaceous origin of angiosperms [55,56,57]. All these studies and ours converge to a consensus that angiosperms did exist in the Jurassic, far earlier than widely accepted [25,26]. Such an outcome of debate is rather expected, since “No angiosperms until the Cretaceous” is a conclusion hard to defend but very easy to falsify.

## 5. Conclusions

More than six decades ago, all pre-Cretaceous records of angiosperms were rejected. Since then, it became a well-accepted idea that there were no angiosperms before the Cretaceous. This idea made the great diversity of angiosperms in the Early Cretaceous as if out of nothing. Recent years have witnessed increasing traces of angiosperms in pre-Cretaceous, recognized by various independent research groups working on fossils as well as molecular clocks, undermining the validity of the idea. In this paper, a new angiosperm *Qingganninginfructus formosa* is described from the Middle Jurassic of Northwest China. The specimens are of infructescences with elongated or round-triangular samaroid fruits, each with a single basal anatropous bitegmic ovule, a short pedicel, and a narrowing or truncated tip. Based on a detailed comparison with known gymnosperms and angiosperms, we assign the present fossils to a new taxon, *Qingganninginfructus formosa*. The discovery of a new genus of angiosperms from the Middle Jurassic not only adds to the diversity of Jurassic angiosperms, but also demonstrates that the angiosperms have flourished during the Jurassic of Europe and Asia. Together with previous fossil evidence, this new knowledge further undermines the long-existing “No angiosperms until the Cretaceous” stereotype. This updated perspective on origin and early history of angiosperms calls for modifications in angiosperm systematics.

## Figures and Tables

**Figure 1 life-13-00819-f001:**
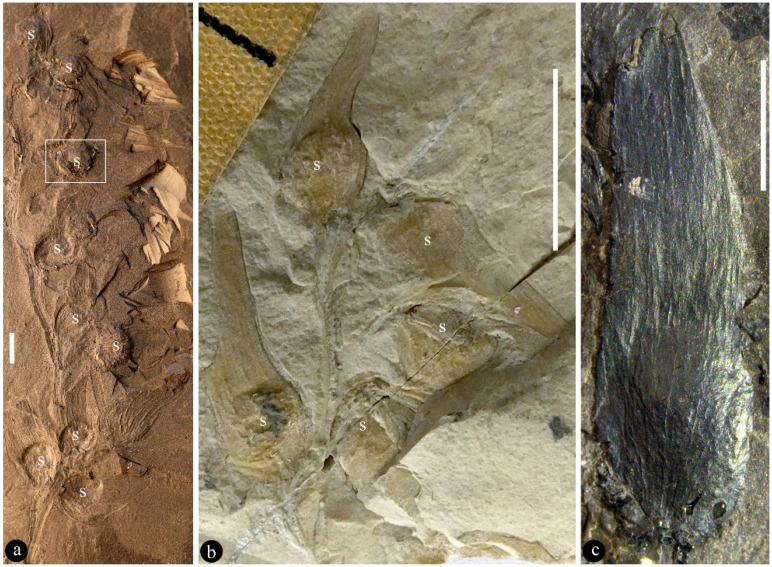
*Qingganninginfructus formosa* and its enclosed ovules. (**a**). An infructescence including several fruits spirally arranged along its rachis. Note the ovule (s) within fruits. LZP-2018-02A. Scale bar = 5 mm. (**b**). A short infructescence including fruits attached to the rachis. Note the ovule (s) within fruits. LZP-1984-58. Scale bar = 5 mm. (**c**). A coalified single fruit with longitudinal striations. Note the bulging basal part suggestive of an ovule/seed (not visible to naked eyes) in the fruit. LZP-2018-09. Scale bar = 5 mm. (**d**). Cross view of the basal portion of the fruit shown in Appendix A, showing two layers of fruit wall (arrows) sandwiching the ovules (s). Shooted following the direction of arrow in Appendix A. LZP-2018-09. Scale bar = 1 mm. (**e**). An exposed ovule in a broken fruit, surrounded and partially covered by fruit wall (arrows), which is magnified from the rectangle area in (**a**). Scale bar = 1 mm. (**f**). An exposed ovule (s) in a broken fruit, partially covered by fruit wall (rectangle). Refer to (**h**). Enlarged from the white rectangle area in Appendix A. Scale bar = 1 mm. (**g**). Side view of a broken fruit showing fruit shape (arrows) and a basal ovule (s). Refer to Appendix A. Enlarged from the black rectangle area in Appendix A. Scale bar = 1 mm. (**h**). Detailed view of the rectangle in (**f**). Note the fruit wall (arrows) over the ovule (s). Scale bar = 1 mm.

**Figure 2 life-13-00819-f002:**
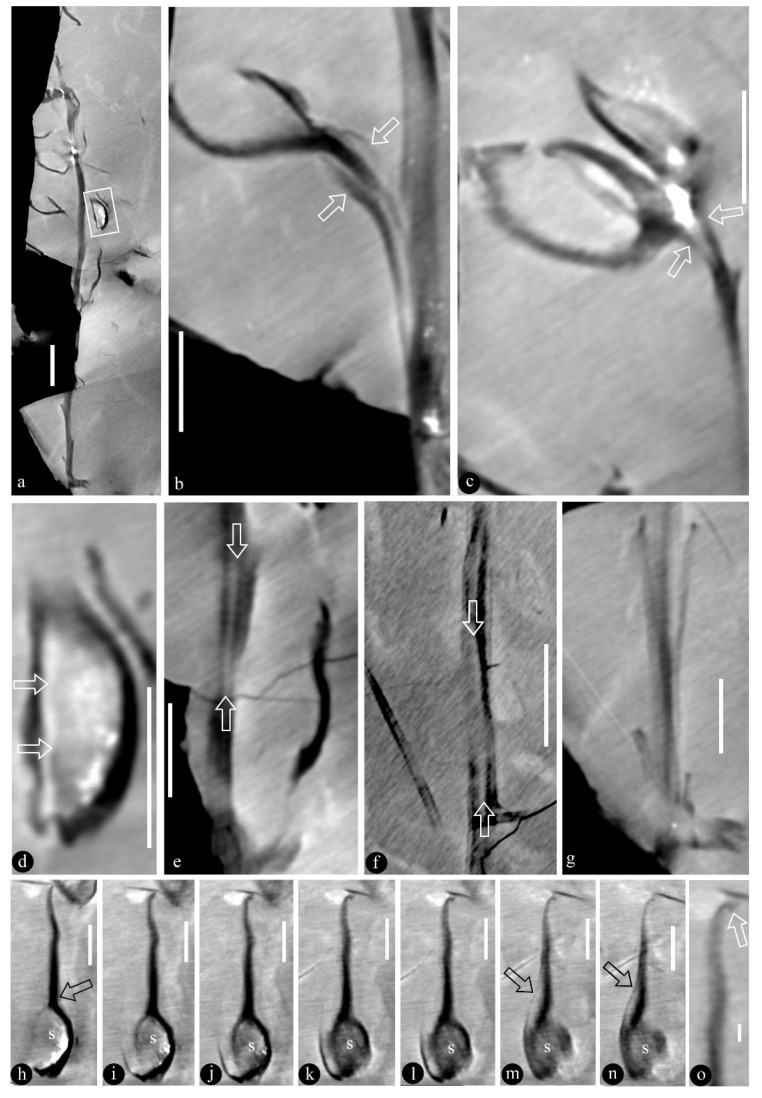
Micro-CT virtual views of *Qingganninginfructus formosa* embedded in a siltstone. LZP-2018-03B. (**a**). An infructescence including several fruits spirally arranged along the rachis. Scale bar = 10 mm. (**b**). A sheathed fruit pedicel. Scale bar = 5 mm. (**c**). Paired fruit pedicels (arrows) attached to the rachis. Scale bar = 5 mm. (**d**). Detailed view of the fruit in the rectangle in (**a**). Note the shape of the ovule (arrows). Scale bar = 5 mm. (**e**). Empty central canal (arrows) in the rachis. Scale bar = 5 mm. (**f**). Empty central canal (arrows) in a closely associated branch. Scale bar = 10 mm. (**g**). Spiral arrangement of leaf stubs (?) along the basal portion of the rachis in (**a**). Scale bar = 5 mm. (**h**–**n**). Serial sections of a fruit showing an ovule (s) inside the fruit (arrows). Scale bar = 5 mm. (**o**). Detailed view showing curved tip (arrow) of the fruit in (**l**). Scale bar = 1 mm.

**Figure 3 life-13-00819-f003:**
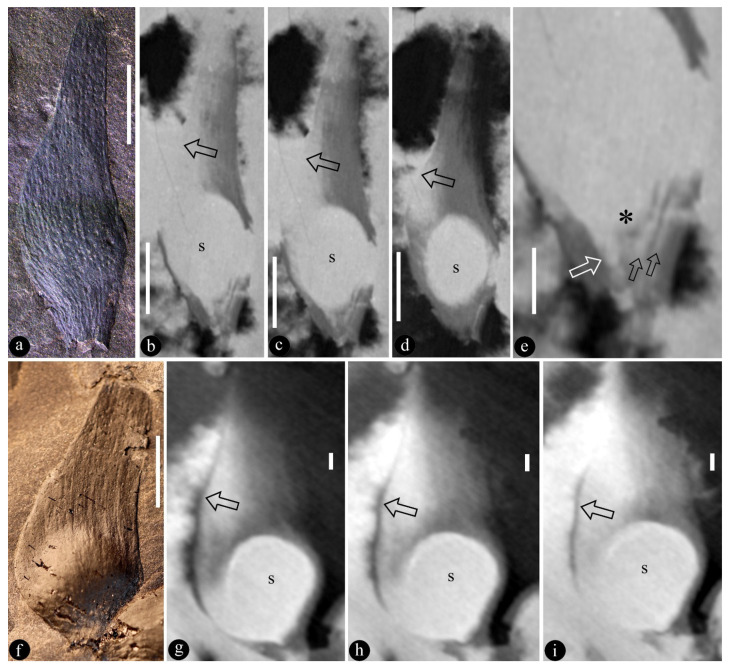
Two isolated fruits with enclosed ovules of *Qingganninginfructus formosa*, revealed by Micro-CT images. (**a**). An isolated fruit with longitudinal striations and bulging basal portion, suggestive of an enclosed ovule in the fruit. LZP-2018-06A. Scale bar = 5 mm. (**b**–**d**). Serial Micro-CT views of the fruit in (**a**), showing ovule (s) in the fruit (arrows). Scale bar = 5 mm. (**e**). Detailed view of (**b**), showing funiculus (white arrow), nucellus (asterisk), two integuments (black arrows), and micropyle between these three. Scale bar = 2 mm. (**f**). Another isolated fruit with longitudinal striations and bulging basal portion, suggestive of an enclosed ovule in the fruit. LZP-2018-03B. Scale bar = 5 mm. (**g**–**i**). Serial sections of a fruit showing a basal ovule (s) in the fruit (arrows) shown in (**f**). Scale bar = 1 mm.

**Figure 4 life-13-00819-f004:**
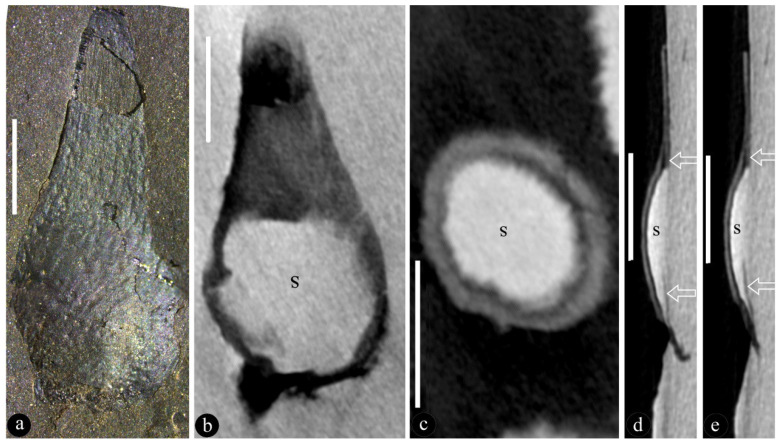
An isolated fruit of *Qingganninginfructus formosa* and its ovule revealed in Micro-CT virtual views. LZP-2018-05A. (**a**). An isolated fruit with longitudinal striations and bulging basal portion, suggestive of an enclosed ovule in the fruit. Scale bar = 5 mm. (**b**). Longitudinal surface section of the fruit in (**a**), showing ovule (s) within. Scale bar = 5 mm. (**c**). Longitudinal surface section of the fruit, showing the ovule (s) surrounded by fruit wall (gray). Scale bar = 5 mm. (**d**,**e**). Longitudinal sections of the fruit showing the ovule (s) between the overlying fruit wall and underlying fruit wall (arrows). Scale bar = 5 mm.

## Data Availability

All data are reported in this paper.

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
