# Peer review of "New Fossil Evidence Suggests That Angiosperms Flourished in the Middle Jurassic"

_life, 2023, doi:10.3390/life13030819_

Round 1
Reviewer 1 Report
My Review
Certainly, this report is extremely important in presenting the early angiosperms in Jurassic. However, there are two major weak areas that should be revised and improved.
The first weakness is on the fruit type. Obviously, samara fruit type is a very important characteristics of angiosperms. However, while the authors recognized this important feature, they mentioned samara only in the abstract and generic diagnosis. They did not describe the fruits using the terms for samara fruit, such as nutlet, wings, long keel, angle, and so on. I would strongly recommend the authors to read some articles related to samara fruits, and then re-describe the fossil material. Here are just a few example articles:
A paper with detailed description of samara fruit parts of Acer: Gabrielyan, Ivan, and Johanna Kovar-Eder. "The genus Acer from the lower/middle Pleistocene Sisian formation, Syunik region, south Armenia." Review of Palaeobotany and Palynology 165.3-4 (2011): 111-134.
Another paper on Acer fruits: Zhu, Hai, and Steven R. Manchester. "Red and silver maples in the Neogene of Western North America: Fossil leaves and samaras of Acer section Rubra." International Journal of Plant Sciences 181.5 (2020): 542-556.
In Acer (maple), the angles between the paired fruits (or say between the long keels) are commonly larger than 180 degrees, to me which is different from that of Qingganninginfructus that has the paired samara with a narrow angle between the two long keels, like a butterfly.
Stigmaphyllon has the wings similar to that of Qingganninginfructus. See: Almeida, R. F., and Andre Marcio Amorim. "Stigmaphyllon caatingicola (Malpighiaceae), a new species from seasonally dry tropical forests in Brazil." Phytotaxa 174.2 (2014): 82-88.
A comparation between the samara fruits of Qingganninginfructus with other fossil and extant angiosperms with samara fruits should be presented in the discussion.
The second major weakness is on the ovule type. The authors described it too briefly, just “The ovule is anatropous and bitegmic (Figs. 3a-b, e).” While either “anatropous” or “bitegmic” is a very important characteristics of angiosperms, each should be demonstrated and described with a lot of details. In Figs. 3b-c, 3e (not 3a-b, e), where is micropyle? Where is funiculus? Since Dr. Wang and Santos have just published a new genus Combina, and discussed about anatropous ovules, the anatropous ovules/seeds really should be demonstrated and described more with details here. Bitegmic ovule is even more important characteristics of angiosperms, so it should be described with more details too. Could you provide the video in the supplementary material?
Should Combina be compared and discussed in the paper as it has anatropous ovules and was compared with Drepanolepis too? See: Santos, A. A., and X. Wang. "Pre-Carpels from the Middle Triassic of Spain. Plants 2022, 11, 2833." (2022).
After these are addressed in the manuscript, I believe the authors would give different discussion and conclusion.
Below are some comparatively minor problematic parts:
Line 13-18. The first sentence initiates a topic that that angiosperms are closely related human beings. Does “closely” mean they are biologically related? If not, how are they related? The word “However” usually leads to a sentence disagree the statement before. Thereafter, “related …. However ….. controversial…… no angiosperms in the Pre-Cretaceous age (are there human beings?) …. But ……to reveal the truth” entangled these sentences together.
Line 18-19: You are documenting a new genus Qingganninginfructus with three species of angiosperms from the Middle Jurassic of Northwest China, using Micro-CT technology. Be simple.
Line 20: “most completely preserved specimen” Are you covering only one specimen? If not how many?
Line 20-21: an infructescence including several samara fruits, each with a single nutlet in the basal part.
Line 21-23: Since the fossils are distinct from all organs of known gymnosperms and angiosperms are literally defined by their enclosed seed, we interpret Qingganninginfructus as a new genus of angiosperms in-cluding three species from the Middle Jurassic of Northwest China.
Line 24-26: The discovery of a new genus of angiosperms from the Middle Jurassic undermines the validity of invalidates the “no angiosperms until the Cretaceous” stereotype conclusion and updates the perspective on the origin and early history of angiosperms.
Line 30:The early history of angiosperms frequently is evidenced commonly glimpsed with by a single or a few fossil specimens.
Line 31: Another improper use of “However” as it disagrees with the previous statement. How about write it positively as “Therefore, the discoveries of the so-far-earliest angiosperm fossil plants Schmeissneria [3,4] and Nanjinganthus [5,6] from the Early Jurassic with over tens or hundreds of specimens are apparently exceptional.”?
Line 32-36: The number of specimens and “the effect” are not comparable things/concepts, but you can say “However, these new discoveries appear not enough to counterbalance shake off the influence of “no angiosperms until the Cretaceous” conclusion [7,8] that has been taught in classrooms in the past six decades. Therefore, it is not surprising that these new discoveries arose controversies [9–11].”
Line 36: Which papers are “these papers” in “although the arguments presented in these papers are flawed and misleading [6]”, [5,6], [7,8], or [9–11]? Are you attacking or defensing?
Line 37: To make reading smoothly, it is better to avoid of changing subjects/standing points among sentences. How about “Obviously, it requires further fossil evidence to expel the ‘no Pre-Cretaceous Angiosperms’ stereotype.”?
Line 40: Please understand the difference between “plant fossils” (for the rock) and “fossil plants (for the plant).” Use them properly, please.
Line 64-65: Change the sentence to be: à We cannot exclude the possibility that future studies on Nathorst’s materials, using modern technology to reveal three-dimensional information, will find enclosed seeds in situ in Nathorst’s materials.
Line 68: “have widespread” à had spread widely
Line 75-82: “Fruits are in pairs or single” à “Fruits are paired or singled samara type.”
Fig. 1a. Should the “seeds” be nutlets? Which one has exposed seeds? Which “seeds” in 1a are magnified to 1e and 1g? They should be marked in Fig 1a.
Fig. 2. Is there a pedicle sheath in 2C as that in 2B?
2e and 2f have hollow peduncle and hollow branch (empty).
Could the leaves in 2G be the sheath as that in 2B? What do the leaves look like?
Fig, 3. Figure 3. Two isolated fruits with enclosed seeds revealed by Micro-CT.
3B-D. Is the so-called “seed” just a hollow chamber, or actually there are something?
3E. Detailed view of Fig. 3b, showing funiculus (white arrow) and two integuments (black arrows). à Label the micropyle. Please provide a video.
3F the right edge appears to be the dorsal side, based on the “venation” type, but in G-I the thick long “line” on the left side should be the long keel on the dorsal side. Are they of the exact same specimen?
Figure 4. An isolated fruit of Qingganninginfructus formosa gen. et sp. nov. and its nutlet revealed through Micro-CT virtual views (4B-E).
4B. Longitudinal surface section view of the nutlet in Fig. 4a. “Surface view” is conflicting with internal view.
4C. Longitudinal surface section view of the nutlet, showing the seed (s) surrounded by fruit wall (gray).
D-E. Longitudinal sections of the fruit showing the seed (s) between the overlying fruit wall and underlying fruit wall (arrows). è Why is the underlying fruit wall so short? Broken?
Supplimentary
Beautiful venation on samara in S3b, S3d, and S4H can be used for comparison.
Nice reticulate venation on nutlets(?) in S3b, S3f has a samara (on the top) with. And reticulate venation at the nutlet part. Could these be used for comparison too?
Paired samara in S4B, S4I, S4J (see my comments about paired samara).
Line 27: Change “a seed (arrow). Fig. 1d is shooted following the direction of the arrow” to be “a seed (arrow) with the seed’s side view (in the direction of the arrow) exhibited in Fig. 1d”?
S6. Did you try to observe/photograph the specimens under fluorescent light?
S7. "showing its triangular shape (?)
Line 31: Add a space before “Scale bar = 5 mm”.
Reviewer 2 Report
The manuscript "Angiosperms flourished in the Middle Jurassic, new fossil evidence suggests" is well written, with relevant information in the introduction, material and methods well explained with the use of modern techniques and the conclusions are supported by the data.
I am not a specialist in paleobotanic nomenclature, but I think the proposed new names need the authors: Qingganninginfructus (Zhang)Wang (or other study authors)gen. nov.
The species need the authors as well to be valid.
Reviewer 3 Report
This is an interesting manuscript that describes a new genus and three species of early angiosperms from the Middle Jurassic of China. The description of the new taxa is based on macromorphology in combination with Micro-CT imaging. This paper contributes to our knowledge of early angiosperms. However, I consider that the manuscript needs to be improved to be published:
1. The argumentation to define the new genus Qingganninginfructus needs to be improved and explained in more detail.
2. The new genus should be compared with other similar fossil angiosperms and gymnosperms genera, apart from Drepanolepis or Schizolepis.
3. I don't see enough morphological differences to establish three diferents species in this new genus. Is not it possible that the new taxa is monospecific having shape variation?
4. I recommend to rephotograph the specimens displayed on Figure 1a, 1b and 1d. Figure S3a-h. Figure S4a-j.
I have also indicated directly on the original submission (seed attached pdf) my comments and suggestions to improve the quality of the manuscript.

Round 2
Reviewer 1 Report
My Review of life-2126830-peer-review-v2
First of all, authors have changed a lot based on my previous comments and suggestions, so the paper quality has improved a lot. There are still one big question to be answered and some minor aspects/parts need to be addressed, as I point out below.
The big question is: are there real seeds in the fossil fruits? While authors claimed the fruits have seeds enclosed inside, they never mentioned about embryo. Also, the two integuments and micropyle found the Micro-CT video indicate the integuments had not fully sealed, and thus no embryo developed yet. And thus, so called “seeds” are neither seeds nor seed coats, but just ovules or integuments. All Micro-CT videos should be carefully examined to see if there is any embryo exist in fruits. If no seeds are found, then all “seeds” in text and figure captions should be changed to be “ovules” or “integuments” properly. Then that could indicate the ovules dropped before seeds are formed as in the case of some orchids, cycads, and ginkgoes. This would not weaken the significance of discovery of the fossil fruits. If it is impossible to use Micro-CT to scan all fruits, then just stated that no embryo has been found in the scanned specimens yet.
The rest problematic aspects/parts are mostly vague or grammatic:
Line 16: “the early history and origin of angiosperms” should be “the origin and early history of angiosperms”, because origin comes first.
Line 20: Flowers à fossil flowers
Line 22: Should “new species of angiosperms” be a new genus? If not, what is the species name?
“Qingganninginfructus formosa gen. et sp. nov.” should be introduced at the first time mentioned here. Then you can use the genus name only, in line 23.
Line 24, 61, 244: “Micro-CT observation reveals” or “Micro-CT reveals” should be modified because Micro-CT itself can neither observe nor reveal anything, it is you guys who found something in the Micro-CT video.
Line 24: “the most completely preserved specimen” in unclear. Do you mean “the best-preserved fossil infructescence has eleven samaroid fruits”?
Line 25-27: Two “are” in a clause cause confusing. Do you mean “Since these fossils are distinct from all organs of known gymnosperms and angiosperms AND THEY are literally defined by their enclosed seed”? If not, please reword the sentence.
Line 28 and 165: indeterminate species should be changed to be unspecified species, because “The indeterminate species are those species which are considered under the category of threatened species.”
Line 41: Please replace “will” with “can”, because “will” implies “not happened yet”.
Line 43: Any reference for the stereotype? Since Scott et al., 1960?
Line 44 and 170: “Reveal” again. The phrase “without revealing criteria” sounds as those authors intentionally hid their criteria. How about using neutral word “giving” rather than “revealing”?
Line 45: Please replace the first “started” with “began”.
Line 47-49: The long sentence with “Although …., despite …..” is confusing. How about delete “Although valuable”? Then the sentence seems clearer.
Line 49: What are “the conclusions”? The conclusions of Ref. 12-14? All their conclusions?
Line 49: “difficulty (of “rarity of fossil specimens”?) cannot be overcome by (increasing the) number of specimens”? This is self-conflicting. How about “Furthermore, even more specimens are found, the …. still exists, because ……”?
Line 53: From which did Scott et al (1960) adopt?
Line 49-54: This long sentence is very hard to follow. It got me lost.
Line 60-61: Please change “This new technology has recently revealed” to be “Using this new technology, Fu et al. (2023) has recently revealed”.
Line 63-64: Change to be: “This successful attempt encourages us to use the Micro-CT to have revealed that some fossil plants of previously reported as Drepanolepis formosa by Zhang et al. [34] actually have fruits with enclosed seeds.”
Do you reexamine your previously-reported fossil specimens or new fossil specimens of a fossil taxon that was previously reported by others? Reference of the “previously documented” should be given here. Delete “reported here” since you formally declared it in line 69.
Line 65-68: Move these two sentences to the end of Introduction, and re-organize the sentences in Line 65-73, please.
Line 77 and 120: Be aware of the difference between peduncle and rachis. Replace “peduncle” with “rachis” please, as the peduncle is the axis below the first flower on an inflorescence.
Line 77 and 131: It is better to change “Fruit single or in twins” to be “Fruit single or in pairs”, to be consistent with that in Line 131, plus “twins” have special biological meaning, as you know.
Line 77: “variable in form” is too vague. Please give any typical forms, such as from form xxxx to form xxxx.
Line 81-83: Ningxia is not a province, but Ningxia Hui Autonomous Region. Think about how to treat these Province/Region properly, please.
Line 105: Does “no information” mean that Santos and Wang did not mention integument at all?
Line 106-107: “distinct from” à How? In what aspects?
Line 127: “Besides that of the genus” à “In addition to the generic diagnosis”?
Line 169-171: Do you mean “Unbelievably, many earlier works on early angiosperms routinely did not give the criteria to define angiosperms”?
Line 179-182: The two sentences are chronologically confusing, logically referring the three papers [18, 2017], [39, 2009], and [40, 2019]. Also, it is normal that some co-authors of an earlier paper can change opinions in a new article. Over-emphasis on this kind of change increases your confronting tones, which I think is unnecessary.
Line 187: What do you mean by “trouble” and which “botanists” are you referring in “Sokoloff et al.’s trouble comes not only from other botanists”?
Line 190-191: I would suggest delete “What on the earth is Bateman’s real idea on the criterion for angiosperms? No knows unless Bateman tells the truth.” “No one knows”? What do you imply by “the truth”? It is better not raise personal, confronting tones in a science paper.
Line 196-197: “To be consistent, the present authors prefer to adopt this criterion to reach the conclusion in this paper” has some misleading and unnecessary words, such as, to be consistent with what? What conclusion? The conclusion of the whole paper [I felt the paper ending here, when read “reach the conclusion in this paper”]? Could it be simplified as “The present authors prefer to adopt this criterion to define angiosperms.”?
Line 231: Does “micropyle between these three” mean “micropyle between the funiculus and the two integuments”? Then, the two integuments are only found on the right side. Right? No integument on the left side? Any interpretation? Remember: in the video, the enclosed linear structure (that should be the integument, not seeds) is found only on one side two (not a complete oval shape).
Because I have heavy teaching, annual evaluation, and other engagements, I will stop here, as I am pushed to complete this review before tomorrow, Saturday. I think I have found and pointed out the most important things, so the authors can correct other similar or related problematic parts, if any exists, of the whole manuscript.
I hope my comments and suggestions can help the authors to improve the manuscript to be an excellent one.
Reviewer 3 Report
The authors have incorporated most of the comments I had on the original manuscript, and I think this revised version is an improvement, and I recommend publication.
Author Response
Dear Reviewer,
We are grateful for your constructive comments and suggestions. These suggestions have undoubtedly helped to improve the manuscript greatly. Thanks for your valuable time and support to us.
Best regards
Yours sincerely
Xin Wang and Lei Han
Round 3
